# The Role of Neuroactive Steroids in Analgesia and Anesthesia: An Interesting Comeback?

**DOI:** 10.3390/biom13111654

**Published:** 2023-11-15

**Authors:** Vesna Jevtovic-Todorovic, Slobodan M. Todorovic

**Affiliations:** Department of Anesthesiology, Anschutz Medical Campus, University of Colorado, Aurora, CO 80045, USA; slobodan.todorovic@cuanschutz.edu

**Keywords:** developmental neurotoxicity, young brain, old brain, attention deficit hyperactivity disorder (ADHD), autism spectrum disorder (ASD), alphaxalone, neuroapoptosis, pain post-surgery

## Abstract

Published evidence over the past few decades suggests that general anesthetics could be neurotoxins especially when administered at the extremes of age. The reported pathology is not only at the morphological level when examined in very young and aged brains, given that, importantly, newly developing evidence suggests a variety of behavioral impairments. Since anesthesia is unavoidable in certain clinical settings, we should consider the development of new anesthetics. A promising and safe solution could be a new family of anesthetics referred to as neuroactive steroids. In this review, we summarize the currently available evidence regarding their anesthetic and analgesic properties.

## 1. Introduction

Over the past three decades, the anesthesia field has been grappling with a real possibility that the currently used general anesthetics (GAs) could be powerful neurotoxins especially at the extremes of age [1,2,3,4,5,6,7,8,9,10,11,12,13,14]. Several animal studies and some human ones suggest that elderly brain functions could also be powerfully impaired after GA administration, especially after prolonged and/or multiple exposure [13,14]. In addition, numerous animal and human reports suggest that anesthesia-induced neuronal demise during the early stages of brain development as assessed by widespread neuroapoptotic cell death leads to on-going disturbances in neuronal connectivity, synaptic plasticity and interneuron dynamics, suggesting that the sequence of detrimental outcomes stems from massive and widespread neuronal cell death [1,2,3,4,5,15,16]. Hence, it would appear that the currently used GAs are potent pharmacological disruptors of many aspects of neuronal development, leaving the ‘surviving’ neurons in a disarrayed milieu as they try to engage in physiological processes and morphological adaptation that are fundamental to normal behavioral development. This would suggest that cell death may not be the most important or even a necessary initial step leading to long-lasting neurodevelopmental abnormalities, but rather, on-going maladaptive changes in neuronal morphology and synapsing could be the culprit when rapidly developing and/or vulnerable neurons are exposed to unphysiological conditions brought on by GAs. As discussed in several reports over the past few years, it appears that long treatment durations or repetitive exposures are not necessary for GAs to cause neuropathological changes within the developing brain [17], although they seem to be important when it comes to the aging brain. Short exposures, single exposures, and even exposures that are subanesthetic are sufficient to permanently alter the developmental trajectory of a vulnerable brain [6,7,8].

Over the past two decades, there has been a steady accumulation of clinical evidence pointing at GAs as being potentially detrimental to the behavioral development of the youngest members of our society. Although there have been several reports suggesting poor outcomes regarding the cognitive impairments [9,10,11,12,18], rapidly accumulating evidence over the past decade has been suggestive of the potential association between early exposure to GAs (before the age of two) and an ADHD diagnosis later in life (as monitored until the age of 19 years) [7,8,17,19,20,21]. Even after adjusting for gestational age, sex, birth weight and comorbid health conditions, it was concluded that early exposure to GAs was associated with a three-fold increase in the risk for the later development of ADHD (starting at the age of eight years) [7,19]. Of note, the same population cohort was shown in an earlier report (2009) to have a two-fold increase in the incidence of developmental delay diagnoses later in life (as monitored until the age of 19 years) [6]. Importantly, both reports regarding an increase in developmental delay and ADHD diagnosis in exposed children were confirmed in another population cohort study published several years later (in 2017) [7]. Very recently, a published meta-analysis that looked at several retrospective cohort studies found a significant link between an early exposure to GAs prior to the age of three and a subsequent diagnosis of ADHD even when adjusted for comorbid health conditions [17]. It is noteworthy that in some important respects, these behavioral human reports are complementary to others in non-human primates [22,23], thus suggesting that although it would be more comfortable to minimize or ignore these findings, we owe it to our children to consider the possible iatrogenic role that GA exposure early in life may play in the development of ADHD later in childhood.

As for ASD, concerns have been reported as well [24,25,26]. A recent European study suggested potentially detrimental effects of early GA exposure based on the fact that there were two-fold increases in the reported incidences of ASD and autism in previously anesthetized children [26]. Although several variables were considered—importantly, health and economic status—the outcomes of the analysis strongly suggested that the earlier the exposure to GAs occurred, the more likely the reported outcomes were suggestive of ASD. Of particular importance for this review is the conclusion of this study indicating that the risk of autism or ASD was more profound when there was no exposure to surgery, indicating that GAs may be an independent risk factor.

In view of this evidence, one must ponder the need for the development of new GAs that are effective and safe. The U.S. Food and Drug Administration (FDA) agrees.

Recently, the FDA issued a black box warning recommending that clinicians should discuss with families the potential neurological harm of long-duration or repetitive exposures to common GAs (i.e., propofol, ketamine, sevoflurane, isoflurane, midazolam, etc.) [27,28]. The FDA made a step further this year by inviting the investigation of safer sedatives and anesthetics for clinical use, with a clear focus on a behavioral follow-up, thus reemphasizing the growing concern and signaling the need for the introduction of promising new sedatives and GAs. Considering that anesthesia is unavoidable in many clinical settings, we are struggling to find ways to assure the well-being of children and our aging patients whilst providing comfort during painful procedures. It appears that a promising and safe solution may not currently be available, i.e., a solution that would meet those needs and, as such, would transform patient lives, with far-reaching benefit to our society. However, a new family of GAs referred to as neuroactive steroids could offer a promising new lead.

## 2. Neuroactive Steroids as Promising Anesthetic Agents

Having a basic four-ring structure, neuroactive steroids could be chemically modified with different reactive groups, creating different analogs and thus giving rise to different pharmacological effects on neurosensory processing and neuronal excitability [14,29]. The most common cellular target of interest in the field of anesthesiology is their positive allosteric modulation (PAM) of GABA-A receptors, since neuroactive steroids that are potent GABA-A PA modulators were reported to be the most potent GAs. In addition to PAM mechanisms of GABA-A receptors that most commonly involve inhibitory ligand-gated channels [30,31], it appeared that other ligand-gated channels as well as voltage-gated ion channels could play an important role in this process. The question, of course, was whether neuroactive steroids with strong PAM of GABA-A activity would have the same neurotoxic potential as the currently used GAs that are also potent PA modulators of GABA-A receptors, which prompted a quest for neuroactive steroids with a different putative target. Due to their role in neuronal excitability and their ability to get activated at low membrane voltage, T-channels became the most interesting target [32,33,34]. They regulate several important aspects of intrinsic neuronal excitability and synaptic neurotransmission as well as modulate pain transmission, sleep–wake cycles and seizure activity. Of particular interest for our research has been a neuroactive steroid analog, 3β-OH [(3β,5β,17β)-3-hydroxyandrostane-17-carbonitrile] (Figure 1), a potent voltage-dependent blocker of T-currents in native neurons without a significant blocking effect on voltage-gated Na^+^ and K^+^ currents, N-type and L-type high-voltage-activated (HVA) VGCCs or recombinant GABA_A_- or NMDA-mediated currents within the range of hypnotic brain concentrations [35]. It is noteworthy that 3β-OH, when compared to allopregnanolone, is not known to have appreciable GABAergic effects. Interestingly, 3β-OH is known to exert some modulatory action on the glutamatergic system. This effect seems to be mediated by AMPA receptors via a decrease in glutamate release.

Although 3β-OH does not have direct effects on GABA-A receptors, our recent study demonstrated that its peripheral metabolite, 3α-OH ((3α,5β,17β)-3-hydroxyandrostane-17-carbonitrile), is a potent PA modulator of GABA-A receptors. The 3α-OH metabolite also has T-channel blocking properties with sex-specific hypnotic and pharmacokinetic effects similar to those of 3β-OH. Interestingly, the peripheral conversion of 3β-OH to 3α-OH is more prominent in adult female than in adult male rats and likely contributes to the stronger hypnotic properties of 3β-OH in females [36]. Although it is not known if similar peripheral metabolic pathways of 3β-OH exist in neonatal rodents, the important question was posed whether 3β-OH would be a safe hypnotic in very young animals. Recently published findings suggested that 3β-OH has a favorable therapeutic index when compared to the intravenous anesthetic ketamine but, unlike ketamine, when used in equipotent doses, a 12-h 3β-OH-induced anesthesia did not cause a significant increase in neuroapoptosis [37]. This led to the conclusion that 3β-OH is an effective hypnotic that does not damage to the young brain. Together with our earlier work suggesting the potent analgesic properties of 3β-OH [32,33,34], this led us to believe that 3β-OH could be a promising new GA for use in a young population.

Although the findings regarding the lack of neurotoxicity are very promising, the relevance to behavioral outcomes remains at the forefront. An initial focus on cognitive development showed that early exposure to 3β-OH during a peak of synaptogenesis did not result in long-term learning deficits in rats examined during adulthood [37]. On the contrary, exposure to equipotent doses of ketamine led to delays in learning and information retention, resulting in a significant gap in learning speed and in the ability to achieve a priori determined learning criteria. We concluded that certain steroid analogues may be safe alternatives to the currently used GAs and should be carefully examined.

Although the PAM of GABA-A function by GA action seems to be associated with developmental neurotoxicity, the question remains whether neuroactive steroids with GABA-A PAM properties could still be safe alternatives, which would suggest intrinsic ‘neuroprotective’ properties of this class of drugs. To address this notion, we investigated two other neuroactive steroid analogs with prominent PAM properties with respect to GABA-A receptors, namely, CDNC24 ([(3α,5α)-3-hydroxy-13,24-cyclo-18,21-dinorchol-22-en-24-ol]) and alphaxalone [(5α-pregnan-3α-ol-11,20-dione)] (Figure 1) [38]. We confirmed that, despite their direct GABA-A PAM properties and unlike commonly used intravenous anesthetics with PAM properties (propofol), they had no neurotoxic properties and were safe for the young rodent brain. The mechanism underlying these seemingly intrinsic neuroprotective properties remains to be confirmed. The initial work appeared to suggest that both CDNC24 and alphaxalone decrease GABA discharge by acting presynaptically [38], an effect that is crucial in balancing the activation of extrasynaptic GABA receptors [39] at clinically relevant concentrations. In contrast, propofol is known to have mainly postsynaptic GABAergic effects resulting in an increase in spontaneous inhibitory postsynaptic GABA currents [38]. In view of this observation, one may propose that the simultaneous blockade of T-type voltage-gated calcium channels (at least in the case of alphaxalone) and a reduction in presynaptic GABA release may be, at least in part, responsible for their hypnotic action that is devoid of damaging effects in young neurons. Considering that 3β-OH and CDNC24 are neurosteroids, their mechanism of action may also be intracellular and/or nuclear in nature.

## 3. Neuroactive Steroids in Clinical Practice

Neuroactive steroids consist of a core structure of 17 carbon atoms shaped into four rings, with A–C being cyclohexane rings, and D being a cyclopentane ring, as can be seen in Figure 1. Different steroid compounds contain different functional groups at each carbon atom which define their respective mechanisms of action. In the analysis of neuroactive steroids’ anesthetic effects, it was suggested that naturally occurring steroids (deoxycorticosterone and progesterone) have the strongest anesthetic effects compared to other tested steroids [40]. We now know that the potency of the sedative/hypnotic and anesthetic effects of neuroactive steroids are, in fact, directly related to their chemical structure. It is important to note that most endogenous and synthetic neuroactive steroids are very lipophilic and, hence, have been modified to increase their water solubility. The first successful synthesis of water-soluble steroids provided hydroxydione (21-hydroxy-5β-pregnane, 3,20-dione), which was used in the clinical setting [41,42].

Having said that, of importance to this review is the fact that the most successful steroid compounds at producing anesthesia are not water-soluble. These compounds were characterized as being 5α- or 5β-reduced by the steroid enzymes 5α- or 5β-reductase. Structurally, they have hydroxy functional groups positioned in the α configuration on carbon number 3 (C3) on the A ring of the steroid structure. This occurs after reduction by 3α-hydroxysteroid dehydrogenase, which is an oxidoreductase that mediates reversible reactions. This chemical conversion of endogenous steroids generates neuroactive steroids with anesthetic properties, such as allopregnanolone. The configuration of the hydrogen groups on C5 plays a lesser role in anesthetic potency, as compounds with a hydrogen group in either the α or the β configuration produced anesthesia [29]. One of the most successful compounds at producing hypnosis and anesthesia is 3α-hydroxy-5α-pregnane-11,20-dione, or alphaxalone.

The introduction of neuroactive steroids in human anesthesia was inauspicious in most generous terms. The first synthetic neuroactive steroid to be used clinically was alphaxalone (3α-hydroxy-5α-pregnane-11,20-dione) (Figure 1), used to induce and maintain general anesthesia in the early 1970s, when it was marketed as Althesin for human use. Interestingly, this first formulation also included alfadolone (at a 3:1 ratio), another neurosteroid with weak anesthetic properties. Although initially considered to be a good GA, the main barrier from the beginning proved to be their lipophilic nature and, hence, their poor solubility. The Althesin formulation included 20% polyoxyethylated castor oil (Cremophor EL) as a solvent. Unfortunately, it was reported very early on (by the mid-1970s) that Althesin caused severe allergic reactions described in numerous cases to be anaphylactic in nature, resulting in severe respiratory and cardiovascular collapse. As a result, Althesin was withdrawn from human use in 1984. Of note, a new alphaxalone formulation called Alfaxan was released in Australia in 2001 and eventually was approved for veterinary use in the United States in 2012 for the induction and maintenance of anesthesia in dogs and cats. Alfaxan contains a non-cremophor vehicle (2-hydroxpropyl-β-cyclodextrin) which does not cause histamine release and, hence, is much less likely to cause any allergic reactions. Although frequently used in animals without significant side effects, it has been reported that 2-hydroxpropyl-β-cyclodextrin can cause diarrhea in humans.

A recent revisit of alfaxalone as a potentially promising GA for human use came with the development of a new formulation called Phaxan, an aqueous solution containing 10 mg/mL of alfaxalone in 13% betadex (7-sulfobutyl ether β cyclodextrin, known by its commercial name of Captisol). Today, Captisol is in at least 13 FDA-approved injectables and in numerous clinical candidates [43,44]. Based on recently published information, it appears that this formulation does not cause allergic reactions, as confirmed in three major databases, Toxline, Medline and EMBase. In preclinical studies with adult patients, Phaxan was shown to be a fast onset–offset IV anesthetic like propofol, but with less effects on cardiovascular depression. In currently published clinical trials focused on assessing the safety, efficacy and quality of the anesthetized state and recovery, it was reported that Phaxan induced fast-onset, short-duration anesthesia similar to the commonly used intravenous anesthetic propofol, with fast cognitive recovery but with less cardiovascular depression or airway obstruction [43]. Unlike propofol, there were no reports of pain on injection. The conclusion was made that in its current formulation, Phaxan is a fast-acting intravenous anesthetic with high therapeutic index in humans.

In a more recent clinical study, Phaxan anesthesia was compared to propofol and propofol-plus-sevoflurane anesthesia in a double-blinded prospective randomized trial focused on patients undergoing hip arthroplasty [45]. When the anesthetic doses in each group of patients were titrated to the same depth of anesthesia (BIS 40-60), it was reported that the alphaxalone-anesthetized patients scored better than the propofol- and sevoflurane-anesthetized ones in cognition tests such as the Grooved Pegboard Test, the Digit Symbol Substitution Test (DSST) and the Mini Mental State Examination (MMSE) for seven days postoperatively. Interestingly, the higher cognition scores were accompanied with higher serum mature brain-derived neurotrophic factor (m-BDNF) levels in the alphaxalone-anesthetized patients when compared to the two other groups, a noteworthy observation, considering that the m-BDNF levels were reported to be decreased in anesthesia-induced neurotoxicity studies in both young and aged brains, where the scarcity of astrocytes was accompanied with a decrease in the levels of secreted m-BDNF. Thus, the following question arises: would it be safe to state that alphaxalone offers neuroprotection by promoting neuronal survival instead of the neuronal demise observed with currently used GAs? This could be an important area of future investigations.

## 4. Neuroactive Steroids as Promising Analgesics

The use of opioids in the operating room and in the clinic has steadily increased over the last decades, making them one of the most commonly prescribed classes of drugs in the U.S.A. This is due to the fact that the currently used GAs are not very effective analgesics in cases of significant surgical stimulation and tissue injury. As a result, we have been using a variety of opioid agents for the management of acute pain in the immediate perioperative period. The problem with this practice has been the fact that opioids exhibit significant side effects, most notably addiction and tolerance. As reported in 2010, over 12 million Americans were registered as being opioid abusers [46,47], resulting in more overdose deaths than heroin and cocaine combined [48]. Other currently available medications have either limited efficacy or serious side effects. For example, regional anesthesia and the use of local anesthetics may not be suitable if early mobility after surgery is desirable. Furthermore, non-specific effects of local anesthetics such as numbness may prevent an early neurological assessment after surgery [49]. Finally, regional anesthesia is contraindicated for many patients who are anticoagulated. Thus, further research into new therapeutic modalities for the treatment of pain and their relationship with opioid analgesics in the perioperative period is warranted. From both mechanistic and translational perspectives, it is important to point out again that different neuroactive steroids that are either PA modulators of neuronal GABA-A receptors and/or inhibitors of Ca_V_3.2 channels may have an effect on one or both of the two major systems involved in pain transmission in the dorsal horn (DH) of the spinal cord and in dorsal root ganglion (DRG) neurons of peripheral sensory nerve fibers.

### 4.1. Voltage-Gated Ca^2+^ Channels (VGCCs) and Nociception

VGCCs are ubiquitous in both central and peripheral neurons and play a major role in shaping action potentials and in controlling cellular excitability and synaptic transmission. It is well established that N-type Ca^2+^ channel blockers have a major function in presynaptic inhibition in the DH of the spinal cord. In clinical studies, the selective N-type channel blocker ziconotide was established to be effective in the treatments of patients with intractable pain [50]. The cloning of the pore-forming α1 subunits of T-channels showed the existence of at least three channel subtypes: G (Ca_V_3.1), H (Ca_V_3.2) and I (Ca_V_3.3) [51]. It is well established that Ca_V_3.2 channels are essential for nociception [52,53,54,55], as also shown by our recent studies in animals with a clinically relevant model of surgical plantar skin incision [56,57].

### 4.2. GABA_A_ Ligand-Gated Channels and Nociception

Under normal physiological conditions, γ-aminobutyric acid (GABA) exerts tonic inhibition of nociceptive neurotransmission between primary afferents and second-order spinothalamic tract neurons [58,59]. Hence, the pharmacological antagonism of spinal GABAergic neurotransmission results in mechanical hypersensitivity similar to that in chronic pain [60]. In cases of axonal nerve injury, baseline GABA activity and GABA levels are reportedly decreased in the DH, leading to a reduction in primary afferent-evoked inhibitory postsynaptic currents (IPSCs) [61,62] and phenotypic changes in GABA_A_ receptors [63]. It is not clear whether the dysregulation of GABA transmission is due to a loss of GABAergic interneurons in the spinal DH, dorsal root ganglia (DRGs) or both. A few other possible mechanisms have been considered, including the lack of an effective uptake and recycling process due to the down-regulation of the GABA transporter GAT-1 levels, which would result in considerable depletion of GABA from its terminals in the DH, without a concomitant loss of GABA neurons [64]. The possible loss in content or activity of the GABA-synthesizing enzymes glutamic acid decarboxylase (GAD) 65 and 67 was also suggested in some studies [65]. It is important to mention that regardless of the mechanism, a significant imbalance of GABA was found in humans with chronic pain due to peripheral nerve injury [66].

Neuroactive steroids are potent modulators of neuronal activity, causing a variety of behavioral and neuroendocrine changes in humans and animals (e.g., general anesthesia, analgesia, cognitive and mood disturbances) [67,68]. It is usually believed that the effects on neurosensory processing and neuronal excitability are primarily mediated by the modulation of GABA-A receptors by steroids such as alphaxalone. However, we found that a neuroactive steroid with a 5α configuration at the steroid A,B ring fusion [(+)-ECN] [(3β,5α,17β)-17-hydroxyestrane-3-carbonitrile] (see structure in Figure 1) is a potent voltage-dependent blocker of T-channels in nociceptive rat DRG neurons and has very little effect on neuronal voltage-gated Na^+^, K^+^, N- and L-type HVA Ca^2+^ channels and glutamate- and GABA-gated channels [32]. Other studies showed that ECN only weakly inhibits recombinant Ca_V_2.3 currents [69]. Furthermore, we showed that the analgesic efficacy of alphaxalone, ECN and related 5α-reduced steroids is correlated with their ability to potentiate GABA_A_-gated currents and/or inhibit T-currents in DRG neurons [34]. We also identified several synthetic 5β-reduced steroid analogues that lack any direct effect on GABA_A_ currents but potently and completely inhibit T-currents in DRG cells and exhibit potent local analgesic effects in vivo [70]. One of the most potent and efficacious steroid analogues in this group, 3β-OH ((3β,5β,17β)-3-hydroxyandrostane-17-carbonitrile) (Figure 1), is a voltage-dependent channel and a selective blocker of T-currents in acutely dissociated DRG cells. Finally, we also identified in our earlier study a compound (CDNC24, (3α,5α)-3-hydroxy-13,24-cyclo-18,21-dinorchol-22-en-24-ol) that has no effect on T-currents in DRG cells but is a potent potentiator of GABA-A currents [34] with anesthetic properties in tadpoles [71] and rat pups [38]. Figure 1 summarizes the chemical structures of different analogues of neuroactive steroids that we extensively used in our studies of nociception.

### 4.3. Sensitization of Pain Responses Following Surgical Skin Incision

It is particularly interesting that our recent studies showed that these neuroactive steroids have different roles in suppressing the sensitization of pain responses caused by surgical incision in rodent models of post-surgical pain. Nociceptors can become hyperexcitable (sensitized) by various mechanisms in the presence of peripheral tissue injury. Individual nociceptors that have been sensitized can be activated by stimuli that previously would not have been intense enough to cause activation (allodynia) as well as by previously noxious stimuli that now produce an even greater sensation of pain (hyperalgesia). The electrophysiological correlates of these altered evoked pain responses include a lowered threshold for nociceptor activation and an increased frequency of firing in response to a suprathreshold stimulus, respectively. Furthermore, clinical symptoms of spontaneous pain can be directly correlated with the spontaneous spike firing in nociceptors [72]. Peripheral sensitization often leads to central sensitization or increased synaptic efficacy and spontaneous activity of DH neurons in the spinal cord [73,74]. Based on preclinical and clinical evidence, various types of evoked and spontaneous pain are commonly reported in the immediate post-surgical period [75,76]. Although those types of painful symptoms are successfully treated with the use of local anesthetics and opioids, it was proved that the chronic use of opioids may not only be ineffective but also result in serious side effects (as discussed previously) including a condition referred to as opioid-induced hyperalgesia [77]. As a result, the current research efforts have been focused on better understanding various types of pathological states marked by pain sensitization (e.g., spontaneous pain and hypersensitivity to mechanical and thermal stimuli). Hence, we became interested in studying the potential anti-hyperalgesic effects of different neuroactive steroids in rodent models of surgical paw incision.

Furthermore, using this clinically relevant postsurgical animal pain model, we found that when given intraperitoneally, 3β-OH alleviated pain in adult rats when the pain was induced by plantar paw surface surgical incision. The neurosteroid 3β-OH was shown to be effective in alleviating pain when used in combination with isoflurane during the plantar surgical incision procedure. This data indicated that this neurosteroid may be used as a supplementary anesthetic, thus reducing the amounts of conventional anesthetics and also the possibility of side effects [57]. In the same study, we reported that 3β-OH effectively ameliorated incision-induced hyperalgesia in rats when administered intrathecally and locally when injected into incised paws. This points to the relatively unique properties of 3β-OH, which exerted dose-dependent hypnosis when injected systemically and reversed hyperalgesia regardless of the route of administration (e.g., peripheral, central or systemic). Our ensuing studies using mouse genetics demonstrated that the Ca_V_3.2 isoform of T-channels is important for the anti-hyperalgesic properties of 3β-OH in the model of surgical paw incision and also contributes to the induction, but not the maintenance, of neurosteroid-induced hypnosis [57]. In contrast, mice with a global deletion of the Ca_V_3.1 isoform of T-channels [78] as well as of a Ca_V_2.3 isoform of HVA VGCCs [79] showed a decreased duration of hypnosis induced by 3β-OH. This data are consistent with the idea that different hypnotic and analgesic effects of neuroactive steroids like 3β-OH may be mediated by the inhibition of different isoforms of neuronal VGCCs. Specifically, it appears that the induction of hypnosis and analgesia is mediated at least in part by the inhibition of the Ca_V_3.2 isoform, and the maintenance of hypnosis by the inhibition of the Ca_V_3.1 and Ca_V_2.3 isoforms.

We also addressed the role of the preemptive use of intrathecal administration of neurosteroids with inhibitory activity on Ca_V_3.2 T-type currents and a potentiating effect on GABA-A currents using a rat paw incision model [80]. We used neurosteroids with distinct effects on GABA-A receptors and/or T-channels such as alphaxalone (combined PA modulator of GABA-A receptors and T-channel inhibitor), ECN (T-channel inhibitor) and CDNC-24 (PA modulator of GABA-A receptors without blocking T-channels in DRG neurons) and compared them with an established analgesic, morphine (an opioid agonist without known effect on either T-channels or GABA-A receptors). We found that alphaxalone and ECN, but not morphine, caused a lasting relief of mechanical hyperalgesia when administered preemptively, whereas morphine provided dose-dependent pain relief only when administered once the pain had developed. In contrast, we found that CDNC24 did not offer any preemptive analgesic benefit when administered intrathecally in this pre-clinical surgical pain model. We concluded that neurosteroids that inhibit Ca_V_3.2 T-currents are effective preemptive analgesics that may offer a promising therapeutic approach to the treatment of post-incisional hypersensitivity.

However, given that many neurosteroids are potent positive modulators of GABA_A_ ligand-gated ion channels, we question if this effect could be exploited for the treatment of pain following surgery. The role of GABA_A_ modulation in both the spinal cord DH and the DRGs of rodents in nociception was previously studied in chronic preclinical pain models; however, similar studies in an acute incisional pain model are lacking. Considering this, our previously published studies showed that the GABAergic system in the DRGs was down-regulated in rats with sciatic crush injury and that the direct DRG application of the GABA_A_ agonist muscimol [81] or of a specific antisense oligonucleotides (ASODNs) targeting the α2 subunit of ligand-gated GABA-A channels [82] completely abolished or worsened, respectively, the development of neuropathic pain post-crush injury. Based on this information, we speculate that the modulation of GABA_A_ receptors in DRG neurons may also be crucial for the development and progression of evoked and spontaneous pain following deep plantar skin incision. Hence, the elevation of GABA activity in DRGs by neurosteroids like CDNC24 may lessen the primary afferent barrage of DH neurons known to occur after nerve injury and in doing so, may provide early protection against central sensitization manifested as pain hypersensitivity, allodynia and spontaneous pain. The direct targeting of DRGs may be achieved in clinical settings using epidural injections of neurosteroids to provide perioperative analgesia. However, the possible analgesic potential of neurosteroids like CDNC24 that are PA modulators of GABA-A receptors remains to be determined in future preclinical studies.

## 5. Conclusions

Collectively, the on-going use of alphaxalone in medical and veterinary practice and the recent promising preclinical studies with 3β-OH and related analogues continue to encourage the future development of synthetic neurosteroids with useful hypnotic/analgesic properties. These preclinical studies strongly suggest that the inhibition of T-type channels by neurosteroids is important for both their hypnotic and analgesic properties. Although intraperitoneal injections of hypnotic doses of 3β-OH alone did not completely suppress the response to painful stimuli in rodents, they turned out to be effective in lowering the required amounts of the potent volatile GAs (e.g., isoflurane) used to induce surgical levels of anesthesia. Furthermore, the use of regional anesthesia using spinal and epidural techniques could provide high concentrations of neurosteroids in the proximity of nociceptors and provide more complete analgesia without transient paralysis or numbness for surgical procedures. We posit that the use of novel neurosteroids that target VGCCs and GABA-A receptors may provide superior perioperative analgesia and potentially diminish the risk for drug addiction resulting from opioid overuse, while providing safe GAs to our youngest patients, as strongly suggested by our recent pre-clinical studies. However, future preclinical and clinical studies are needed to establish the potential use of neurosteroids and other drugs that inhibit T-channels as effective and safer anesthetics for intravenous administration and as analgesics in the perioperative period.

## Figures and Tables

**Figure 1 biomolecules-13-01654-f001:**
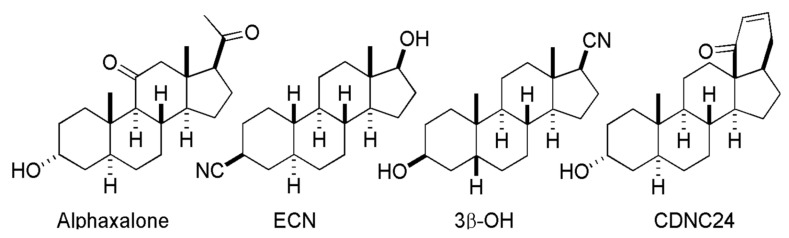
Structures of neuroactive steroids used in our studies.

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
