# Peer review of "The Role of Neuroactive Steroids in Analgesia and Anesthesia: An Interesting Comeback?"

_biomolecules, 2023, doi:10.3390/biom13111654_

Round 1

Reviewer 1 Report

Comments and Suggestions for Authors

The manuscript is well written and will be a timely addition to the literature. I have a few minor suggestions.

Page 1, line 15: Define ADHD and ASD.

Page 2, line 44: Have the deleterious effects mentioned been documented at short exposures with human relevant doses and in stable physiological preparations?

Page 3, line 133: What is the enzyme that converts 3beta-OH to 3alfa? If you know that, you can determine if that enzyme has been found in developing animals and humans.

Page 4, line 164: Awkward sentence structure.

Page 5, line 210: Need a comma after formulation.

Page 5, line 240-243: This sentence needs a reference.

Page 7, line 329: Need a comma after chronically.

Page 8, line 387: delete “is that”.

Author Response

Page 1, line 15: Define ADHD and ASD.  Suggested definitions are now included.

Page 2, line 44: Have the deleterious effects mentioned been documented at short exposures with human relevant doses and in stable physiological preparations?  Yes, please see the reference that is now included (Ing et. al., 2021)

Page 3, line 133: What is the enzyme that converts 3beta-OH to 3alfa? If you know that, you can determine if that enzyme has been found in developing animals and humans. The enzyme that converts 3beta-OH to 3alpha is 3alpha-hydroxysteroid dehydrogenase and is present in animals and humans.  This is now clarified in newly added paragraph in section 3. 

Page 4, line 164: Awkward sentence structure.  This sentence has been revised.

Page 5, line 210: Need a comma after formulation.  Comma inserted. 

Page 5, line 240-243: This sentence needs a reference.  The reference is included as suggested on page 6 line 271

Page 7, line 329: Need a comma after chronically.  Comma inserted.

Page 8, line 387: delete “is that”.  Deleted.

Reviewer 2 Report

Comments and Suggestions for Authors

The review submitted by Vesna Jevtovic-Todorovic and Slobodan M. Todorovic describes the perspectives on the potential use of neuroactive steroids as anesthetic and analgesic agents. Overall, the review is well-written and will be of interest to a wide range of readers. However, in my opinion, there is a weak side to the presented manuscript. It would be worth adding several separate paragraphs devoted to the issues of synthesis, metabolism, and classification of endogenous neuroactive steroids, which synthetic derivatives are discussed in this review. In this regard, it is worthwhile to consider the molecular mechanisms underlying the effects of neuroactive steroids in more detail. In particular, it is worth mentioning whether the effects on the described voltage-gated channels or receptors are a consequence of the direct interaction of steroids or an indirect effect through signaling cascades associated with receptors or channels. Adding this information will make the review more complete and exciting.

Author Response

It would be worth adding several separate paragraphs devoted to the issues of synthesis, metabolism, and classification of endogenous neuroactive steroids, which synthetic derivatives are discussed in this review.  Please see newly added paragraphs at pages 4-5, lines 180-204.

In this regard, it is worthwhile to consider the molecular mechanisms underlying the effects of neuroactive steroids in more detail. In particular, it is worth mentioning whether the effects on the described voltage-gated channels or receptors are a consequence of the direct interaction of steroids or an indirect effect through signaling cascades associated with receptors or channels. Adding this information will make the review more complete and exciting.  The effect on described voltage-gated channels or receptors is a consequence of a direct interaction with steroids.  This is clarified on page 4 lines 153-178.